# Genomic and Metabolic Insights into Two Novel *Thiothrix* Species from Enhanced Biological Phosphorus Removal Systems

**DOI:** 10.3390/microorganisms8122030

**Published:** 2020-12-18

**Authors:** Andrey V. Mardanov, Eugeny V. Gruzdev, Dmitry D. Smolyakov, Tatyana S. Rudenko, Alexey V. Beletsky, Maria V. Gureeva, Nikita D. Markov, Yulia Yu. Berestovskaya, Nikolai V. Pimenov, Nikolai V. Ravin, Margarita Yu. Grabovich

**Affiliations:** 1Research Center of Biotechnology of the Russian Academy of Sciences, Institute of Bioengineering, 119071 Moscow, Russia; mardanov@biengi.ac.ru (A.V.M.); gruevg@yandex.ru (E.V.G.); mortu@yandex.ru (A.V.B.); nravin@biengi.ac.ru (N.V.R.); 2Department of Biochemistry and Cell Physiology, Voronezh State University, 394018 Voronezh, Russia; songolifreya@mail.ru (D.D.S.); ipigun6292@gmail.com (T.S.R.); maryorl@mail.ru (M.V.G.); nikita.markov257@mail.ru (N.D.M.); 3Winogradsky Institute of Microbiology, Research Center of Biotechnology of the Russian Academy of Sciences, 117312 Moscow, Russia; jberestovskaja@mail.ru (Y.Y.B.); npimenov@mail.ru (N.V.P.)

**Keywords:** metagenome-assembled genome, biological phosphorus removal, colorless sulfur bacteria, *Thiothrix*

## Abstract

Two metagenome-assembled genomes (MAGs), obtained from laboratory-scale enhanced biological phosphorus removal bioreactors, were analyzed. The values of 16S rRNA gene sequence identity, average nucleotide identity, and average amino acid identity indicated that these genomes, designated as RT and SSD2, represented two novel species within the genus *Thiothrix*, ‘*Candidatus* Thiothrix moscowensis’ and ‘*Candidatus* Thiothrix singaporensis’. A complete set of genes for the tricarboxylic acid cycle and electron transport chain indicates a respiratory type of metabolism. A notable feature of RT and SSD2, as well as other *Thiothrix* species, is the presence of a flavin adenine dinucleotide (FAD)-dependent malate:quinone oxidoreductase instead of nicotinamide adenine dinucleotide (NAD)-dependent malate dehydrogenase. Both MAGs contained genes for CO_2_ assimilation through the Calvin–Benson–Bassam cycle; sulfide oxidation (*sqr*, *fccAB*), sulfur oxidation (rDsr complex), direct (*soeABC*) and indirect (*aprBA*, *sat*) sulfite oxidation, and the branched Sox pathway (SoxAXBYZ) of thiosulfate oxidation to sulfur and sulfate. All these features indicate a chemoorganoheterotrophic, chemolithoautotrophic, and chemolithoheterotrophic lifestyle. Both MAGs comprise genes for nitrate reductase and NO-reductase, while SSD2 also contains genes for nitrite reductase. The presence of polyphosphate kinase and exopolyphosphatase suggests that RT and SSD2 could accumulate and degrade polyhosphates during the oxic-anoxic growth cycle in the bioreactors, such as typical phosphate-accumulating microorganisms.

## 1. Introduction

Filamentous colorless sulfur bacteria (FCSB) have been a classic object of microbiology since the time of S.N. Winogradsky, who used them in the research that culminated in the discovery of chemosynthesis. Nevertheless, the genomics and metabolic potential of the FCSB from the genus *Thiothrix* remain poorly understood [1,2,3]. The bacteria retained in the genus after the last taxonomic revision [4] include five species: *T. nivea*, *T. fructosivorans*, *T. unzii*, *T. caldifontis* and *T. lacustris*. Despite the fact that the genomes of *T. lacustris* (GCF_000621325.1), *T. caldifontis* (GCF_900107695.1), and *T. nivea* (GCF_000260135.1) are available from databases, comprehensive analysis of the main metabolic pathways of sulfur, carbon, and nitrogen has not been reported. In addition, 25 metagenome-assembled genomes (MAGs) obtained from metagenomes of natural hydrogen sulfide biotopes are currently available. Two of these genomes were attributed to certain species, *T. nivea* DOLJORAL78_51_81 [5,6] and *T. lacustris* A8 [7], and 23 others were designated as *Thiothrix* sp. or uncultured *Thiothrix* sp. [8,9,10].

Due to the feature of energy metabolism, where hydrogen sulfide is the main donor of electrons during chemosynthesis, sulfur springs were considered to be a traditional habitat for *Thiothrix*, where filamentous bacteria often form massive fouling. There are reports of intense growth of *Thiothrix* in wastewater rich in reduced sulfur compounds [11,12,13,14]. An active growth of *Thiothrix* in wastewater treatment plants leads to the formation of fouling that can disrupt the operation of the entire system, degrading the quality of the wastewater and increasing the treatment time [15,16]. On the other hand, the ability of *Thiothrix* from activated sludge to accumulate polyphosphates and participate in the removal of phosphorus from wastewater was found along with other polyphosphate accumulating organisms (PAOs) [17,18,19], such as ‘*Candidatus* Accumulibacter phosphatis’, *Tetrasphaera* [20,21], and *Halomonas* [22].

The present study aims to analyse of genomic and metabolic potential of two recently obtained *Thiothrix* genomes—*Thiothrix* sp. RT and *Thiothrix* sp. SSD2, recovered from metagenomes of laboratory-scale enhanced biological phosphorus removal bioreactors (EBPR). Based on the phylogenetic analysis of MAGs, we propose two new species, ‘*Candidatus* Thiothrix moscovensis’ sp.nov. and ‘*Candidatus* Thiothrix singaporensis’ sp.nov.

## 2. Materials and Methods

### 2.1. PAO-Enriched Laboratory Culture for Thiothrix sp. RT

Enrichment and cultivation of the PAO microbial community in a lab-scale bioreactor operated at Research Center of Biotechnology (Moscow, Russia) was reported earlier [23]. Briefly, a 15 L sequencing batch laboratory bioreactor, inoculated with an activated sludge from Luberetskya wastewater treatment plant (Moscow, Russia), was supplied with the following medium: 670 mg/L CH_3_COONa·3H_2_O, 50 mg/L NH_4_Cl, 54.3 mg/L KH_2_PO_4_, 56.7 mg/L Na_2_HPO_4_, and 8.6 mg/L yeast extract. The bioreactor was operated at room temperature in a cyclic mode. A 6 h cycle of cultivation included the stages of filling the bioreactor with fresh medium (30 min), anaerobic growth (110 min), aerobic growth (180 min), precipitation of activated sludge (30 min), and discharge of the spent medium (10 min). Under these conditions, the PAO bioreactor stably removed 75–80% of phosphorus from the medium [23].

### 2.2. Metagenome Sequencing and Assembly of Thiothrix sp. RT

The total DNA was extracted from 1 g of sludge sample using a DNeasy PowerSoil DNA isolation kit (Qiagen, Hilden, Germany). A total of about 1.5 μg of DNA was obtained. A metagenomic DNA sample was sequenced using the Illumina (Illumina, San Diego, CA, USA) platform. The TruSeq DNA library was prepared using NEBNext^®^ Ultra™ II DNA Library Prep Kit (New England Biolabs, Hitchin, UK) and sequenced on the Illumina HiSeq2500 instrument (250 nt single-end reads). Adapter sequences and low-quality reads were removed using Cutadapt [24] and Sickle (https://github.com/najoshi/sickle accessed on 10 November 2020), respectively. A total of about 50 million high-quality reads were obtained. The reads were de novo assembled into contigs using MetaSPAdes v. 3.7.1 [25]. The contigs were binned into MAGs using MetaBAT v. 2.12.1 [26]. Completeness of the MAGs and their possible contamination were estimated using the CheckM v. 1.0.13 tool [27]. The taxonomic assignment of the obtained MAGs was performed using the Genome Taxonomy Database Toolkit (GTDB-Tk) v. 1.1.1 [28] and Genome Taxonomy Database (GTDB) [29].

Gene search and annotation were performed using the RAST server 2.0 [30] followed by manual correction by the searches of predicted protein sequences against the National Center for Biotechnology Information (NCBI) databases.

### 2.3. Genome of Thiothrix sp. SSD2

MAG of *Thiothrix* sp. SSD2 was obtained from sludge from a full-scale wastewater treatment plant (PUB, Singapore) and assembled in a single contig with a total length of 4,541,730 [31]. The information about bioreactor operating conditions, samples preparation, DNA isolation, metagenome sequencing and assembly has been published [31]. Genome annotation revealed 16S rRNA gene, 43 tRNA genes and 4097 potential protein-coding genes [31]. For the present study, the *Thiothrix* sp. SSD2 genome was obtained from NCBI GenBank (CP059265).

### 2.4. Phylogenetic Analysis of MAGs

Average nucleotide identity (ANI) was calculated using an online resource (https://www.ezbiocloud.net/tools/ani) based on the OrthoANIu algorithm, using USEARCH [32]. Average amino acid identity (AAI) between the genomes was determined using the aai.rb script from the enveomics collection [33]. Digital DNA-DNA hybridization (dDDH) calculation was performed using the GGDC online platform (https://ggdc.dsmz.de/ggdc.php# accessed on 10 November 2020).

### 2.5. Nucleotide Sequence Accession Number

The annotated genome sequence of *Thiothrix* sp. RT bacterium was submitted to the NCBI GenBank database and is accessible under the number JAEHNX000000000.

## 3. Results

### 3.1. Assembly of the Genome of New Member of the Genus Thiothrix from EBPR Bioreactor

To obtain MAGs of microbial community members, we sequenced the metagenome of activated sludge from of the PAO bioreactor operated at Research Center of Biotechnology (Moscow, Russia) using Illumina technique. Analysis of the taxonomic affiliation of the obtained MAGs showed that one of them, designated *Thiothrix* sp. RT, belongs to the genus *Thiothrix*. Microscopic analysis of microorganisms from the PAO bioreactor also indicates the presence of cells morphologically similar to cultivated members of the genus *Thiothrix*: cells form rosettes, gonidia, and sheath (Figure 1).

The assembled *Thiothrix* sp. RT MAG consisted of 78 contigs (N50 size of 65,012 bp) with a total length of 3,693,042 bp; the CheckM-estimated completeness of this MAG as 99.8% with 0.92% possible contamination. Genome annotation revealed 16S rRNA gene, 38 tRNA genes, and 3645 potential protein-coding genes. The *Thiothrix* sp. RT 16S rRNA gene shares 97.9% sequence identity with *Thiothrix nivea* DSM 5205 and more than 99.5% identity with several environmental 16S rRNA gene sequences detected in activated sludge [34].

### 3.2. Phylogenetic Analysis of MAGs

Besides MAG of *Thiothrix* sp. RT, only one genome of *Thiothrix* sp. from EBPR bioreactors, *Thiothrix* sp. SSD2, is available [31]. Here we compared these genomes with each other and with genomes of known species of the genus *Thiothrix*, which have all been isolated from natural sulfidic environments.

The identity level of 16S rRNA genes of *Thiothrix* sp. RT and *Thiothrix* sp. SSD2 with genes of other members of the genus *Thiothrix* varies from 93.0 to 97.9%, while the 16S rRNAs of all described representatives of the genus are 93.3-98.9% identical to each other (Appendix A
Appendix A). These values suggest assignment of *Thiothrix* sp. RT and *Thiothrix* sp. SSD2 to novel species within the same genus [35].

All species remaining in the genus *Thiothrix* after the last taxonomic revision [4] form two phylogenetic clusters on the tree, reconstructed on the basis of the nucleotide sequences of the 16S rRNA gene (Figure 2).

The first cluster includes *Thiothrix* sp. RT and *Thiothrix* sp. SSD2 together with *Thiothrix nivea* DSM 5205 and *Thiothrix unzii* A1. The second cluster includes *T. lacustris* BL^T^, *T. caldifontis* G1^T^, and *T. fructosivorans* Q^T^.

The AAI values between *Thiothrix* sp. RT and other *Thiothrix* genomes were from 74.0 to 76.9%, and 74.0–78.5% AAI was detected between *Thiothrix* sp. SSD2 and other species (Figure 3). These values are consistent with the assignment of *Thiothrix* sp. RT and *Thiothrix* sp. SSD2 to new species of the genus *Thiothrix* according to the thresholds proposed for genome-based taxonomic delineation [36]. Both of the two new genomes could also be assigned to novel species according to their ANI (<80.2%) and DDH (<30.2%) values (Appendix A). The obtained values of AAI, ANI and dDDH are below the threshold for species delineation [37].

Based on results of phylogenomic analysis, we propose classifying strains RT and SSD2 as ‘*Candidatus* Thiothrix moscovensis’ sp.nov. and ‘*Candidatus* Thiothrix singaporensis’ sp.nov., respectively. ‘*Ca.* Thiothrix moscovensis’ sp.nov. (mos.co.ven’sis N.L. fem. adj. moscovensis pertaining to the city of Moscow, Russia, where the EBPR bioreactor used to obtain this genome operated). ‘*Ca.* Thiothrix singaporensis’ sp.nov. (sin.ga.po.ren´sis N.L. fem. adj. singaporensis pertaining to the city of Singapore, Singapore, where the EBPR bioreactor used to obtain this genome operated).

### 3.3. Comparative Analysis of Metabolic Pathways

#### 3.3.1. Dissimilatory Metabolism of Sulfur Compounds

Members of the genus *Thiothrix* are known representatives of sulfur-oxidizing prokaryotes. Dissimilatory sulfur metabolism of the genus *Thiothrix* is represented by various sulfur-oxidizing complexes involved in the oxidation of hydrogen sulfide, thiosulfate, elemental sulfur, and sulfite.

Genes that encode enzymes involved in the oxidation of sulfide to elemental sulfur, sulfide:quinone oxidoreductase (SQR) and flavocytochrome c-sulfide dehydrogenase (FCSD) were identified in *Thiothrix* sp. RT and *Thiothrix* sp. SSD2.

The *sqrA*, *sqrF*, and *fccAB* genes were found in the genomes of *Thiothrix* sp. RT and *Thiothrix* sp. SSD2. The *sqrD* gene, phylogenetically close to *sqrF*, was additionally revealed in *Thiothrix* sp. SSD2 (Table 1, Appendix A).

For all members of the genus, including *Thiothrix* sp. RT and *Thiothrix* sp. SSD2, the presence of genes of the SOX multienzyme system that oxidizes thiosulfate was revealed (Table 1, Appendix A). A feature of the genus is the absence of genes for the dimeric protein SoxCD. The oxidation of thiosulfate is carried out by the branched Sox pathway (the SoxAXBYZ enzymatic complex), which oxidizes thiosulfate to sulfate and elemental sulfur, accumulated inside the cells.

Elemental sulfur can be further oxidized to sulfite using the reverse dissimilatory sulfite reductase rDsrAB. In *Thiothrix* sp. RT and *Thiothrix* sp. SSD2 genes of this pathway form one cluster consisting of 14 genes (*dsrABCEFHLMKJOPRS*) and 12 genes (*dsrABCEFHLMKJOP*), respectively (Table 1, Appendix A). An accessory gene *dsrR* was not found in the genome of *Thiothrix* sp. SSD2, while *Thiothrix* sp. RT contains all genes of the *dsr* cluster.

Genomes of *Thiothrix* sp. RT and *Thiothrix* sp. SSD2 contains genes for membrane-bound cytoplasmic sulfite:quinone oxidoreductase SoeABC, which directly oxidizes sulfite to sulfate. Genes for indirect oxidation of sulfite to sulfate (the *aprAB* genes encoding APS reductase (AprAB) and the sat gene encoding ATP sulfurylase (Sat)) were also found. Oxidation of sulfite to sulfate occurs via the APS intermediate.

Thus, it can be assumed that *Thiothrix* sp. RT and *Thiothrix* sp. SSD2, like the well-known pure cultures of the genus *Thiothrix*, have pathways for the oxidation of reduced sulfur compounds, which provide the oxidation of hydrogen sulfide, elemental sulfur, sulfite, and thiosulfate.

#### 3.3.2. Autotrophic CO_2_ Assimilation

All genes encoding enzymes enabling autotrophic assimilation of CO_2_ through the Calvin–Benson–Bassham cycle (except for the sedoheptulose-1,7-bisphosphatase) were found in all members of the genus *Thiothrix*, as well as in *Thiothrix* sp. RT and *Thiothrix* sp. SSD2 (Table 1, Appendix A). It should be noted that the sedoheptulose-1,7-bisphosphatase activity in bacteria is often catalyzed by fructose-1,6-bisphosphatase with dual sugar specificity [38,39]. A key Calvin–Benson–Bassham cycle enzyme, RuBisCo, is represented by three types (IAc, IAq, II) in *Thiothrix* sp. RT and *Thiothrix* sp. SSD2. Another key enzyme of the Calvin–Benson–Bassham cycle, phosphoribulokinase, is encoded by two copies of the *prk* gene, which are phylogenetically distant from each other, which is typical for *Thiothrix* species.

#### 3.3.3. Assimilation of N_2_

Genomes of *Thiothrix* sp. RT and *Thiothrix* sp. SSD2 contain almost a complete set of *nif* genes required for N_2_ fixation in *nifASVB1XX2YENQVWMHDKZT* and *nifASVB1XX2B2ENQV WMHDKZTO* operons. The composition of the genes of accessory nitrogenase subunits differed slightly. Genes for NifB2 which, along with NifB1, regulates the nitrogenase complex [40] and NifO which involved in synthesis of the iron-molybdenum cofactor of nitrogenase [41] were not found in the genome of *Thiothrix* sp. RT. The *nifY* gene, which promotes the maturation of the enzyme through its binding to the MoFe cofactor, was not found in the genome of *Thiothrix* sp. SSD2.

#### 3.3.4. Dissimilatory Nitrate Reduction

Genes for the NarGHI membrane complex involved in the dissimilatory reduction of nitrate to nitrite were found in genomes of *Thiothrix* sp. RT and *Thiothrix* sp. SSD2. Genes *nirBD* that encode the NirBD enzyme involved in the removal of the toxic NO_2_^-^ by assimilatory reduction to NH_4_^+^ were found in the genomes of all members of the genus *Thiothrix*, including *Thiothrix* sp. RT and *Thiothrix* sp. SSD2. This chain is realized in almost all representatives of the genus *Thiothrix*, with the exception of *T. nivea*, in which instead of *narGHI*, the periplasmic nitrate reductase encoded by *napAB* was found (Table 1, Appendix A).

The presence of the *nirS* gene for dissimilatory reduction of NO_2_^−^ to N_2_O was shown for *Thiothrix* sp. SSD2, unlike *Thiothrix* sp. RT. The genomes of both bacteria contain the *cnorBC* genes for respiratory nitric oxide reductase that reduce NO to N_2_O. The *nosZ* gene is missing in both genomes, like in other representatives of the genus *Thiothrix*. The presence of genes for dissimilatory nitrate reduction suggests the ability of *Thiothrix* sp. RT and *Thiothrix* sp. SSD2 to perform anaerobic respiration, where nitrate and nitrous oxide can function as terminal electron acceptors. *Thiothrix* sp. SSD2 was also predicted to reduce nitrite. We demonstrated previously the ability to perform anaerobic respiration in the presence of NO for *T. lacustris* AS, *T. caldifontis*, and *T. unzii* [42].

#### 3.3.5. Phosphorus Accumulation

The accumulation of polyphosphate (poly-P) granules is known for members of the genus *Thiothrix* [17,43]. Both *Thiothrix* sp. RT and *Thiothrix* sp. SSD2 contained genes for polyphosphate kinase PPK (*ppk*), the main enzyme that catalyzes the transfer of ATP (*ppk1*) or GTP (*ppk2*) to the active site of the protein during the synthesis of the poly-P chain [44]. The *ppk1* gene encoding PPK1 was found in the genomes of *Thiothrix* sp. RT and *Thiothrix* sp. SSD2. Since this reaction is reversible, in the case of PPK1, it could be used to make ATP from poly-P [44]. The gene for the exopolyphosphatase EPP (*epp*), which catalyzes the hydrolysis of inorganic phosphorus from the polyphosphate chain, was also found in the genomes of *Thiothrix* sp. RT and *Thiothrix* sp. SSD2. Genes for PPK and EPP are also present in other members of the genus *Thiothrix*.

Thus, both enzymes, PPK1 and EPP, can regulate the accumulation and degradation of polyphosphates in the cell. Genes *phoURB*, *pstSACB* which encodes systems for the regulation and transfer of inorganic phosphorus into the cell were also found in the genomes of *Thiothrix* sp. SSD2 and *Thiothrix* sp. RT. Along with participation in biosynthetic processes, polyphosphates are probably used for temporal ATP storage, which is realized through the activity of polyphosphate kinase PPK1 and exopolyphosphatase EPP in members of the genus *Thiothrix*.

#### 3.3.6. Central Metabolic Pathways

Genome analysis of *Thiothrix* sp. RT and *Thiothrix* sp. SSD2 revealed the presence of a complete set of genes for the Embden–Meyerhof (EM) pathway, the tricarboxylic acid (TCA) cycle, and the glyoxylate cycle (Appendix A).

TCA cycle has a remarkable feature in the known species of the genus *Thiothrix*, as well as in *Thiothrix* sp. SSD2 and *Thiothrix* sp. RT, —the absence of the *mdh* gene of the classic cytoplasmic nicotinamide adenine dinucleotide (NAD)-dependent malate dehydrogenase, which is found in the vast majority of prokaryotes. Instead, the mqo gene encoding membrane-bound FAD-dependent malate:quinone oxidoreductase was found in the genomes of representatives of the genus *Thiothrix*, including *Thiothrix* sp. RT and *Thiothrix* sp. SSD2.

Genomes of *Thiothrix* sp. RT and *Thiothrix* sp. SSD2 include all the genes of the EM pathway. The gene *pckA* for ATP-dependent phosphoenolpyruvate (PEP)-carboxykinase, which decarboxylates oxaloacetate to form pyruvate, was also revealed in the genomes of *Thiothrix* sp. SSD2 and *Thiothrix* sp. RT. The genomes contain genes for pyruvate carboxylase (*pyc*) and decarboxylating NADP-dependent malate dehydrogenase (*maeB*). The two latter enzymes likely shunt the reaction of malate conversion to oxaloacetate in the TCA cycle and glyoxylate cycle in order to ensure the production of sufficient oxaloacetate for gluconeogenesis and amino acid biosynthesis.

The release to constructive metabolism from the TCA cycle occurs at the level of oxaloacetate and 2-oxoglutarate. Oxaloacetate is involved in aspartate synthesis with the participation of aspartate aminotransferase (*aspB*), while the conversion of 2-oxoglutarate is carried out by glutamate synthase (*glt*) to glutamate.

#### 3.3.7. Respiratory Electron Transport Chain

Genomes of *Thiothrix* sp. RT and *Thiothrix* sp. SSD2 encode all the major components of the electron transport chain required to generate energy through oxidative phosphorylation. Complex I, NADH:quinone oxidoreductase, is encoded by the genes forming the *nuoABCDEFGHIJKLM* cluster and separate *nuoN* gene in *Thiothrix* sp. SSD2. The complex is encoded by two gene clusters, *nuoACDEFGHIJLM* and *nuoBKN*, in *Thiothrix* sp. RT.

The succinate dehydrogenase II complex includes a cytochrome b_556_ subunit (*sdhC*), an iron-sulfur protein (*sdhB*), a flavoprotein subunit (*sdhA*), a hydrophobic membrane anchor protein (*sdhD*). The genes form a single *sdhABCD* cluster in the genome of *Thiothrix* sp. RT, and are located in two separate loci, *sdhBCD* and *sdhA*, in *Thiothrix* sp. SSD2.

Complex III, ubiquinol-cytochrome *c* reductase, consists of an iron-sulfur subunit (encoded by the *rpi1* gene), a cytochrome *b* subunit (encoded by the *cytB* gene), and a cytochrome *c* 1 subunit (encoded by the *cyc1* gene).

Complex IV is represented by five types of oxidases: six cbb_3_ type subunits (encoded by *ccoNOPGQS*), six cox subunits (encoded by *coxABCD11,15*), and four cytochrome d subunits of ubiquinol oxidase (encoded by *cydABCDX*). Conversion of ADP to ATP occurs via F-type ATP synthetase, encoded at the *atpABCDEGH* cluster, with a separate gene for the F-type ATP synthetase B subunit, *atpF*. The phosphates required for ATP synthetase to function are supplied by inorganic pyrophosphatase, encoded by the *ppa* gene, and polyphosphate kinase, encoded by *ppk*.

Overall, genome analysis of *Thiothrix* sp. SSD2 and *Thiothrix* sp. RT did not reveal any important differences in the composition of genes for metabolic pathways from other representatives of the genus *Thiothrix*. The scheme of potential metabolic pathways of representatives of the genus *Thiothrix*, based on our genome-based reconstructions, is shown in Figure 4.

## 4. Discussion

Phylogenetic analysis of *Thiothrix* sp. RT and *Thiothrix* sp. SSD2 and other members of the genus *Thiothrix* supported classification of the two new strains as new species ‘*Candidatus* Thiothrix moscovensis’ sp.nov. and ‘*Candidatus* Thiothrix singaporensis’ sp.nov., respectively.

The results of the analysis of *Thiothrix* sp. RT and *Thiothrix* sp. SSD2 genomes show that the studied organisms carry out the respiratory type of metabolism, and that they are facultative aerobes and facultative heterotrophs. Complete oxidation of organic substrates can occur through the TCA pathway with the formation of a transmembrane proton gradient, which is further used by ATP synthase for the synthesis of ATP. *Thiothrix* sp. RT and *Thiothrix* sp. SSD2 are potentially capable of assimilation of molecular nitrogen, anaerobic respiration on nitrates and incomplete denitrification.

Autotrophic growth in the presence of reduced sulfur compounds (hydrogen sulfide, elemental sulfur, sulfite, and thiosulfate) with the participation of enzymatic systems of dissimilatory sulfur metabolism is possible for representatives of the genus *Thiothrix*, including *Thiothrix* sp. RT and *Thiothrix* sp. SSD2, which ensure the lithotrophic growth of these organisms. A feature of thiosulfate oxidation is the presence of a branched Sox pathway (the SoxAXBYZ enzymatic complex) and the absence of *soxCD* genes, which leads to the oxidation of thiosulfate to sulfates and elemental sulfur.

A notable feature of *Thiothrix* sp. RT and *Thiothrix* sp. SSD2, as well as other representatives of the genus *Thiothrix*, is the absence of the classical cytoplasmic NAD-dependent malate dehydrogenase, which is found in the vast majority of prokaryotes. Instead, the *mqo* gene encoding a FAD-dependent malate:quinone oxidoreductase was found in the genomes of two MAGs, as well as in the genomes of other known representatives of the genus *Thiothrix*. The presence of only MQO, as an enzyme capable of directly oxidizing malate to oxaloacetate in TCA and in the glyoxylate cycle, is quite rare in prokaryotes [45]. In terms of energy potential (Gibbs free energy, Δ G ° ‘), this enzyme is much more efficient than MDH and, at the same time, its activity increases more than four times in the presence of acetate [45,46].

The two EBPR systems analyzed used a batch reactor with alternating aerobic and anaerobic conditions with acetate as a carbon and energy source. The fact that we managed to assemble MAGs representing new species of the genus *Thiothrix* may also be due to the fact that the samples were taken from EBPR systems. During organoheterotrophic growth, the presence of MQO in these organisms gave an advantage when growing on acetate relative to organisms with classical MDH. This capacity probably facilitated enrichment of *Thiothrix* species in the reactors, as well as their ability to accumulate and degrade polyphosphates as typical phosphate-accumulating microorganisms. It should be noted that the growth of a number of members of the genus *Thiothrix* (*T. lacustris*, *T. caldifontis*) is preferable under chemolithoheterotrophic conditions (cell biomass increases by 2–3 times). With batch cultivation under organoheterotrophic conditions other than the EBPR system, growth is ineffective and is accompanied by a slight increase in biomass, by 20–30% (our unpublished data).

*Thiothrix* usually grows in nature in organic-poor hydrogen sulfide biotopes or in organic-rich wastewater, but in the presence of sulfide. In this case, bacteria use a lithoautotrophic or lithoheterotrophic type of metabolism.

Periodic changes in the growth conditions realized in the EBPR system probably provided an advantage for the enrichment of new *Thiothrix* species that were not previously identified in natural biotopes and wastewater. There is now sufficient evidence that *Thiothrix* can participate in the removal of inorganic phosphorus [17,18,19]. Analysis of MAGs obtained from EBPR system metagenomes will probably be useful not only for identifying new *Thiothrix* species, but also for creating optimal conditions for the growth of *Thiothrix* as phosphate accumulating organisms in the EBPR system. In particular, the addition of nitrate could probably stimulate the growth of *Thiothrix* in the EBPR bioreactors, since they are able to survive anaerobic conditions by switching to anaerobic respiration on nitrates. The addition of acetate to the culture medium promotes the induction of MQO activity involved in the conversion of malate to oxaloacetate. This process is vital because oxaloacetate is an important precursor of gluconeogenesis and amino acid synthesis of the aspartate branch.

## Figures and Tables

**Figure 1 microorganisms-08-02030-f001:**
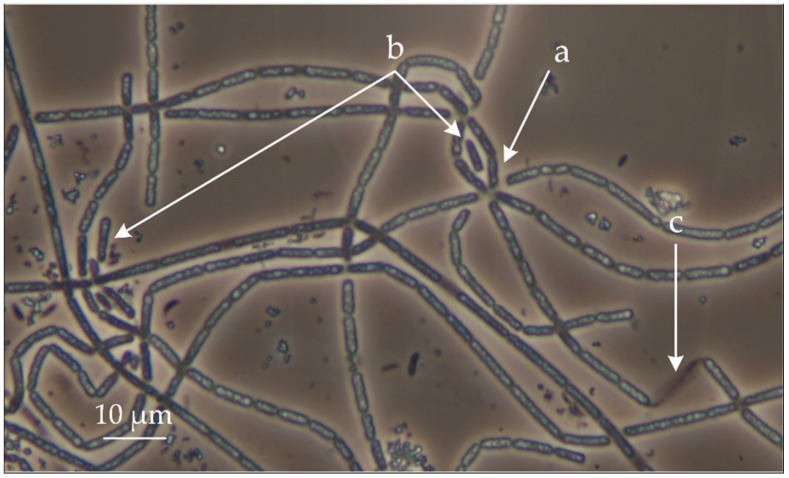
*Thiothrix* sp. as part of a phosphate-accumulating microbial community. Arrows show cells forming rosettes (**a**); detached cells that serve for reproduction (gonidia) (**b**); a sheath (**c**). Cell morphology was observed by using an Olympus CX 41 microscope equipped with a phase-contrast device.

**Figure 2 microorganisms-08-02030-f002:**
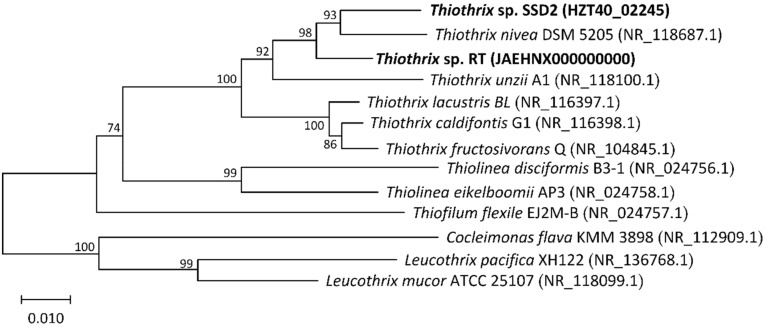
Phylogenetic tree of 16S rRNA gene sequences reconstructed by the neighbor-joining method showing the position of novel strains of *Thiothrix* sp. RT and *Thiothrix* sp. SSD2. The sequences of bacteria from genera Leucothrix and Cocleimonas are included as an outgroup. The percentage of replicate trees in which the associated taxa clustered together in the bootstrap test (2000 replicates) are shown next to the branches. Bar, 0.01 substitutions per nucleotide position.

**Figure 3 microorganisms-08-02030-f003:**
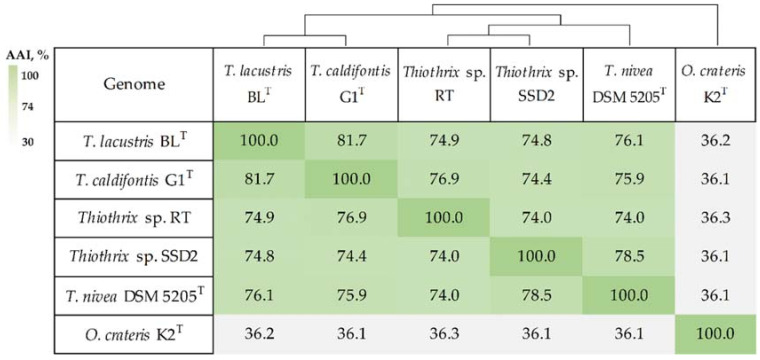
Heatmap of pairwise amino acid identity (AAI) values in the genus *Thiothrix* using *Oceanispirochaeta crateris* K2^T^ as an outgroup.

**Figure 4 microorganisms-08-02030-f004:**
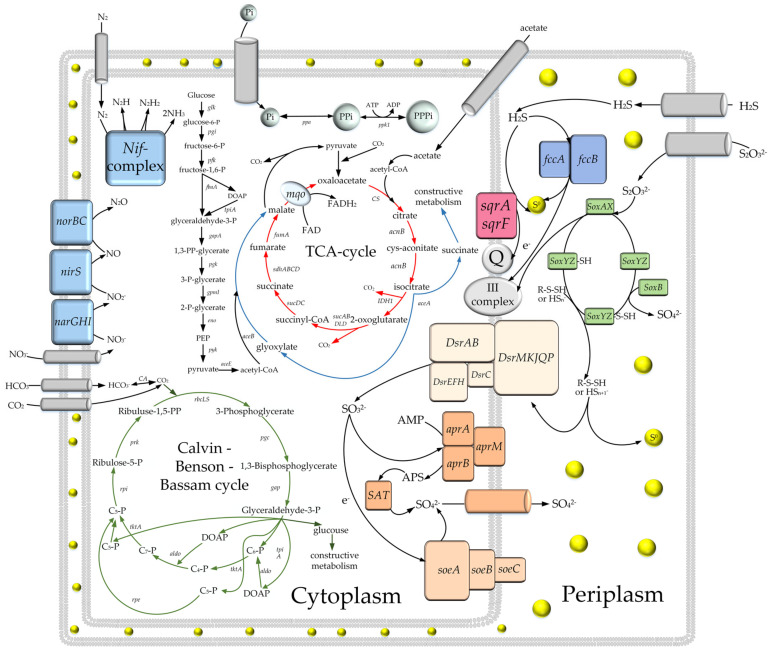
A hypothetical metabolic scheme for *Thiothrix* sp. RT and *Thiothrix* sp. SSD2 reconstructed from the received MAGs. The *nirS* gene encoding nitrite reductase was not found in the *Thiothrix* sp. RT genome.

**Table 1 microorganisms-08-02030-t001:** Genes for nitrogen, sulfur, phosphorus metabolism and key enzymes of the Calvin–Benson–Bassham cycle.

Genes of Metabolic Pathways	*T. nivea* DSM 5205^T^	*T. lacustris* BL^T^	*T. caldifontis* G1^T^	*Thiothrix* sp. RT	*Thiothrix* sp. SSD2
Molecular nitrogen fixation	*nifDKH*	*-*	*nifDKH*	*nifDKH*	*nifDKH*
Dissimilatory nitrate reduction	*-*	*narGHI*	*narGHI, nirS, cnorBC*,	*narGHI,* *cnorBC,*	*narGHI, nirS, cnorBC*
Dissimilatory reduction of NO_3_^−^ to NO_2_^−^, nitrate reductase from the Nap family	*napAB*	*-*	*-*	*-*	*-*
Assimilatory reduction of NO_2_^−^ to NH_4_^+^	*nirBD*	*nirBD,*	*nirBD,*	*nirBD,*	*nirBD*
Dissimilatory sulfur metabolism	*soeABC,* *soxAXBYZ*, *aprAB, sat*, *dsr, sqr*, *fccAB*	*soeABC*, *soxAXBYZ*, *aprAB, sat*, *dsr*, *sqr*, *fccAB*	*soeABC*, *soxAXBYZ*, *aprAB, sat*, *dsr sqr*, *fccAB*	*soeABC*, *soxAXBZY*, *aprAB*, *sat*, *dsr*, *sqr*, *fccAB*	*soeABC*, *soxAXBYZ*, *aprAB*, *sat*, *dsr*, *sqr*, *fccAB*
Calvin–Benson–Bassam cycle	IAc, IAq, II, *prk*(2)	IAq, II, *prk*(2)	IAc, IAq, II, *prk*(2)	IAc, IAq, II, *prk*(2)	IAc, IAq, II, *prk*(2)
Phosphorus metabolism	*phoURB*, *ptsSACB*, *ppk*, *epp*	*phoURB*, *ptsSACB*, *ppk*, *epp*	*phoURB*, *ptsSACB*, *ppk*, *epp*	*phoURB*, *ptsSACB*, *ppk*, *epp*	*phoURB*, *ptsSACB*, *ppk*, *epp*
FAD-dependent malate: quinone oxidoreductase (EC 1.1.5.4)	*mqo*	*mqo*	*mqo*	*mqo*	*mqo*
Malate dehydrogenase (oxaloacetate–decarboxylating) (NADP) (EC 1.1.1.40)	*maeB*	*maeB*	*maeB*	*maeB*	*maeB*

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
