# Peer review of "Genomic and Metabolic Insights into Two Novel Thiothrix Species from Enhanced Biological Phosphorus Removal Systems"

_microorganisms, 2020, doi:10.3390/microorganisms8122030_

Round 1

Reviewer 1 Report

The manuscript describes characterization of two Thiothrix strains possibly contributing to remove phosphorus in a laboratory-scale bioreactor, using metagenome techniques. The metagenomic methods are reasonable and the comparative analysis of genes involved in metabolic pathways in Thiothrix strains are meaningful to understand their ecological niches. The authors intend to propose two provisional (Candidatus) species based on the genomic analysis. Figs 2 and 3 show that it is possible to distinguish the two provisional species from variedly recognized species by gene analysis. Unfortunately, morphological features of the provisional species are not defined. Thiothrix strains are expected to grow forming filaments through longitudinal clumping of cylindraceous cells and the line of cells (filaments) are either sheathed or not. I am not sure that the filaments, cells, and sheaths shown in Fig. 1 (especially Fig. 1c) are those of the provisional species belong to the ginus Thiothrix. The micrographs are confusing because the images are more like Sphaerotilus or Leptothrix than Thiothrix. Introduction of fluorescence in situ hybridization technique using specific proves is recommended to eliminate a feeling of strangeness and to reveal morphological features of the provisional species.

Further comments:

  1. “slimy”; it is not easy to predict an object is slimy or not by phase-contrast microscopy. Deletion of “slimy” is recommended.
  2. “gonidia”; “cells” instead of “gonidia” is reasonable.
  3. “poly-beta-hydroxybutyrate” in Fig. 1 caption; I think it is difficult to detect poly-beta-hydroxybutyrate with methylene blue.
  4. “MAG Thiothrix sp. RT”, “Thiothrix sp. RT MAG”, ”Thiothrix sp. RT” are mixed in the manuscript.
  5. Some of the scientific names (excluding the Candidatus names) are not italicized.

Author Response

Dear Reviewer,

Thank you very much for your questions and recommendations. All revisions was highlighted using the "Track Changes" function in Microsoft Word.

Point 1: The manuscript describes characterization of two Thiothrix strains possibly contributing to remove phosphorus in a laboratory-scale bioreactor, using metagenome techniques. The metagenomic methods are reasonable and the comparative analysis of genes involved in metabolic pathways in Thiothrix strains are meaningful to understand their ecological niches. The authors intend to propose two provisional (Candidatus) species based on the genomic analysis. Figs 2 and 3 show that it is possible to distinguish the two provisional species from variedly recognized species by gene analysis. Unfortunately, morphological features of the provisional species are not defined. Thiothrix strains are expected to grow forming filaments through longitudinal clumping of cylindraceous cells and the line of cells (filaments) are either sheathed or not. I am not sure that the filaments, cells, and sheaths shown in Fig. 1 (especially Fig. 1c) are those of the provisional species belong to the ginus Thiothrix. The micrographs are confusing because the images are more like Sphaerotilus or Leptothrix than Thiothrix.

RE: Thank you for your comment. Indeed, members of the genus Leptothrix were present in the bioreactor (ref 23) according to the 16S rRNA gene profiling, but their relative abundance was 10 fold lower than of Thiothrix species. We agree that bacteria from Fig. 1a and 1c are very similar with bacteria from the Sphaerotilus-Leptothrix group. In order not to mislead readers, we have removed these photos. However, the belonging of filaments from photo 1b (in line 124) to the genus Thiothrix is indicated by the following characteristic features: (1) rosettes formation; (2) cells that serve for reproduction.

Point 2: Introduction of fluorescence in situ hybridization technique using specific proves is recommended to eliminate a feeling of strangeness and to reveal morphological features of the provisional species.

RE: Unfortunately, at the moment we cannot carry out fluorescence in situ hybridization, since there were no samples left from which we sequenced metagenome and assembled MAG.

Further comments:

  1. “slimy”; it is not easy to predict an object is slimy or not by phase-contrast microscopy. Deletion of “slimy” is recommended.

RE: We removed photo 1c from the text, since it does not clearly characterize the morphology of cells of the Thiothrix group. Accordingly, we also removed the caption to this figure from the text (in line 124).

  1. “gonidia”; “cells” instead of “gonidia” is reasonable.

RE: In the caption to Figure 1, we added a definition to the word gonidia in line 126.

  1. “poly-beta-hydroxybutyrate” in Fig. 1 caption; I think it is difficult to detect poly-beta-hydroxybutyrate with methylene blue.

RE: Thank you for this remark. We agree that methylene blue is used in staining for different purposes and it is difficult to detect poly-beta-hydroxybutyrate with it. Fig. 1d is not informative, so we deleted it from the article.

  1. “MAG Thiothrix RT”, “Thiothrixsp. RT MAG”, ”Thiothrix sp. RT” are mixed in the manuscript.

RE:  ‘Thiothrix sp. RT’ is the organism name while MAG of Thiothrix sp. RT or Thiothrix sp. RT MAG refers to its genome. We corrected several confusing instances in the revised text (in lines 129, 133).

  1. Some of the scientific names (excluding the Candidatus names) are not italicized.

RE: We italicized all names as recommended.

Sincerely yours,

Margarita Grabovich

Reviewer 2 Report

1-The staining procedure  for  intracellular inclusions ( fig 1-Intracellular phosphate (methylene blue staining) and poly-β-hydroxybutyrate granules in cells)  should be described, or a citation should be indicated. .Moreover, in fig .1 I can not see any 2-metachromatic staining…

Ther are NO original results in this paper on Biological  Phosphorus Removal  besides the title is

 Genomic and Metabolic Insights into Two Novel  Thiothrix Species From Enhanced Biological  Phosphorus Removal Systems.

At Materials and methods 2.1. PAO-enriched laboratory culture for Thiothrix sp. RT there is the specification that Under these conditions, the PAO bioreactor stably removed 75–80% of phosphorus  from the medium. The results concerning  biological phosphorus removal shoud be presented in the section Results.

3- Their  conclusion Overall, genome analysis of Thiothrix sp. SSD2 and Thiothrix sp. RT did not reveal any  important differences in the composition of genes for metabolic pathways from other representatives  of the genus Thiothrix (Figure 4) is correct , so , the authors should  clearly indicated if figure 4  is original or not.  

Author Response

Dear Reviewer,
Thank you very much for your questions and recommendations. All revisions was highlighted using the "Track Changes" function in Microsoft Word.

Point 1: The staining procedure  for  intracellular inclusions ( fig 1-Intracellular phosphate (methylene blue staining) and poly-β-hydroxybutyrate granules in cells)  should be described, or a citation should be indicated. .Moreover, in fig .1 I can not see any 2-metachromatic staining…

RE: Thank you for this remark. We agree that Fig. 1 is not very informative and raises a lot of questions. In order not to mislead readers, we have removed Figures 1a, 1c and 1d and kept only Fig 1b where cells were not stained. Since our main task was taxonomic description of new Thiothrix species we suppose that the absence of a figure of stained cells with phosphorus inclusions would not affect the main findings reported in our article.

Point 2: Ther are NO original results in this paper on Biological  Phosphorus Removal  besides the title is  Genomic and Metabolic Insights into Two Novel  Thiothrix Species From Enhanced Biological  Phosphorus Removal Systems.

At Materials and methods 2.1. PAO-enriched laboratory culture for Thiothrix sp. RT there is the specification that Under these conditions, the PAO bioreactor stably removed 75–80% of phosphorus  from the medium. The results concerning  biological phosphorus removal shoud be presented in the section Results.

RE: In this work we did not studied phosphorous removal process. The PAO bioreactor (where Thiothrix sp RT was found) has been described earlier (reference 23). We cannot present these data in our Results. Genome sequence of the second Thiothrix sp. (SSD2) was obtained from GenBank and analyzed in this paper. The purpose of this work was to analyze genomes of new Thiothrix species from unusual habitats, - PAO bioreactors, in order to reveal their metabolic potential and taxonomic status. All Thiothrix species described to date have been found in natural hydrogen sulfide - rich biotopes such as sulfidic springs. We found two new species and got some insights into their functional role in the PAP bioreactors.

Point 3: Their  conclusion Overall, genome analysis of Thiothrix sp. SSD2 and Thiothrix sp. RT did not reveal any  important differences in the composition of genes for metabolic pathways from other representatives  of the genus Thiothrix (Figure 4) is correct , so , the authors should  clearly indicated if figure 4  is original or not. 

RE: Figure 4 is original. We added appropriate sentence at the end of Results (in line 290-291).

Point 4: In Abstract the authors stated that

 The values of 16S rRNA gene sequence 19 identity, average nucleotide identity and average amino acid identity indicated that these genomes,  designated as RT and SSD2, represented two novel species within the genus Thiothrix, ‘Candidatus  Thiothrix moscowensis’ and ‘Candidatus Thiothrix singaporensis’. And it seems appropriate to develop and  clearly argue this sentence within Disscutions.

RE: Thanks for the recommendation. In the Results section we argue for this conclusion. In the Discussion section we have added a proposal arising from the results (in line 297-299).

Sincerely yours,

Margarita Grabovich

Round 2

Reviewer 1 Report

The revised manuscript is acceptable for publication. I would like to recommend inserting space between “sp.” and “nov.” throughout the manuscript.